# How Knowledge about Stem Cells Influences Attitudes towards Breastfeeding: Case Study of Polish Women

**DOI:** 10.3390/ijerph18052382

**Published:** 2021-03-01

**Authors:** Malgorzata Witkowska-Zimny, Dorota Majczyna

**Affiliations:** Department of Biophysics and Human Physiology, Medical University of Warsaw, Zwirki i Wigury 61 St, 02-091 Warsaw, Poland; dorota.majczyna01@gmail.com

**Keywords:** breastfeeding promotion, breast milk, stem cells, knowledge

## Abstract

Maternal breastfeeding is associated with multiple health benefits, both for the child and the mother. Since breastfeeding rates are declining, finding new, future-oriented strategies to strengthen and support mothers’ positive attitudes towards breastfeeding need to be developed. In this paper, we describe how information about the presence of stem cells in breast milk can influence the willingness to breastfeed in the population of Polish pregnant women. A cross-sectional study involving a group of 150 pregnant women was conducted to assess the correlation between their knowledge about stem cells and their attitude towards breastfeeding. Among the respondents, only 6% claimed that they did not know anything about stem cells, but general knowledge about stem cells in the research group was poor. The survey results indicated that city residence, university degree, maternal experience and advanced pregnancy correlated with higher general knowledge regarding stem cells. Most respondents (77.3%) had no knowledge regarding the presence of stem cells in breast milk. Approximately two-thirds of mothers with earlier negative breastfeeding experience declared that information about the presence of stem cells in breast milk could have influenced the decision to continue and extend the time of breastfeeding. Hence highlighting the presence of stem cells in breast milk can be used to encourage breastfeeding as a unique activity.

## 1. Introduction

The World Health Organization (WHO) and the American Academy of Pediatrics recommend exclusive breastfeeding (i.e., no other fluids or solids) for the first 6 months of life and then continued breastfeeding combined with the intake of solid foods for the next 2 years or as long as the mother and baby desire [1,2]. In many countries, these guidelines are implemented and adapted according to local conditions. However, breastfeeding provides much more than just nutrition. Breastfed infants have a reduced risk for life-threatening infections, diarrhoea, obesity and several non-communicable diseases in later life, while mothers themselves benefit from a reduced risk for breast and ovarian cancer [3]. Despite its established benefits, breastfeeding is no longer a norm in many communities and, with the introduction of breast milk substitutes, became less common in high-income countries in the 20th century [4]. Research shows that exclusive breastfeeding (EBF) rates have stagnated over the last two decades. Currently, only 37% of infants below the age of 6 months are exclusively breastfed [2,3]. Poland has a high rate of initiating breastfeeding after birth (97%) with a subsequent rapid decline in exclusive breastfeeding (43.5% at 2 months, 28.9% at 4 months and 4% at 6 months) [5]. Some European and Asian countries had low breastfeeding prevalence in the late 20th century but have seen improved breastfeeding rates since then [2]. In Ireland and the United Kingdom, breastfeeding rates remain the lowest among comparable countries in the 2020s. According to the French Institute for Health Surveillance, most French mothers breastfeed for an average of 17 weeks [6]. The French Institute for Public Health Surveillance reports that less than 25% of French mothers breastfeed for 6 months or more. One of six global nutrition targets of the WHO for 2025 is to increase the rate of exclusive breastfeeding in the first 6 months of life up to at least 50% [2]. The mother’s decision to breastfeed is influenced by multiple complex factors at the individual, family, health system, and societal level [7]. Several studies have identified breastfeeding self-efficacy, knowledge, and attitudes as important determinants of breastfeeding outcomes as far as its initiation, duration and exclusivity are concerned [8,9,10]. A woman’s personal values, beliefs and perceptions influence her breastfeeding experience and success [11]. There is clear evidence that besides professional support for breastfeeding mothers, knowledge and the education of pregnant/breastfeeding women raise breastfeeding rates [12,13]. However, mothers do not always receive information about the results of the newest lactation research, which can impact their breastfeeding decisions.

Stem cells are unspecialized cells found in many tissues that have the unique features of unlimited self-renewal and, at the same time, they can transform into various types of mature cells [14]. As they multiply and differentiate, they are a constant source of new body cells in the process of growth and development, but also in the regeneration of damaged tissues. Due to the above-mentioned properties, stem cells have become the subject of extensive research and numerous discussions [15,16]. Scientific reports confirm their use in the treatment of many diseases with previously unfavourable prognoses [17,18,19]. The best-known use of stem cells is hematopoietic stem cell therapy, routinely used in the treatment of bone marrow neoplasms [20]. The umbilical cord blood and perinatal tissues, such as Wharton’s jelly within the umbilical cord, the placenta or the amniotic fluid, are a non-invasive source of stem cells to be used for therapeutic purposes [21,22]. Hence, the birth of a baby offers a unique opportunity to harvest tissues that are rich in stem cells. Thanks to the process of cryopreservation immediately after birth, following a non-invasive procedure that does not raise ethical concerns, perinatal stem cells can be successfully isolated, stored and used in future therapies [23]. Most medical organizations, including the American Academy of Pediatrics (AAP), recommend a public donation of cord blood whenever possible [24]. Information about perinatal tissue banking should be provided to women before pregnancy or at least as soon as possible before delivery. From 2019, new standards for perinatal care have been imposed on healthcare providers (mainly midwives) in Poland, including the obligation to inform women about the right to deposit umbilical cord tissue or its blood stem cells [25].

Confirmation of the presence of stem cells in human milk is one of the most promising and surprising discoveries of the last few years [26]. Groundbreaking research has revealed that maternal cell transfer involves not only immune protective cells but also stem cells via breastfeeding. The work of Professor Peter Hartmann’s team from the University of Western Australia has set a new direction for research into breast milk and lactation [27], raising questions about the nature of these cells, their potential, significance and practical application. Experiments in animal models have shown that stem cells derived from the mother’s milk of a mouse not only transfer to the baby but also survive in the acidic environment of the digestive system and retain the ability to migrate to various organs. Post-mortem examination of young mice showed the presence of labelled cells in many organs: the heart, the brain, the thymus and the pancreas [28]. This suggests the ability of breast milk stem cells to differentiate into the tissues in which they will end up in the baby’s body. It is worth noting that some of the milk-derived cells did not express stem cell-specific markers with time, which implies that they have diversified and become an integral part of the new host tissues [29]. Undeniably, stem cells are of great interest to researchers who have high hopes for their use in modern medicine. Regardless of the research conducted in laboratories, which is focused on the biology and potential of these cells, information about the presence of stem cells in breast milk should be widely used in the education and promotion of breastfeeding.

The aim of the study was to assess the impact of women’s knowledge about stem cells on their attitudes and breastfeeding practices. To the best of our knowledge, there have been no studies exploring breastfeeding attitudes and information about the presence of stem cells in breast milk. In this paper, we describe how information about stem cells abundance in breast milk can relate to the intention to practice breastfeeding.

## 2. Materials and Methods

A cross-sectional study was conducted to ascertain the correlation between the knowledge and opinions of pregnant women about stem cells and their attitudes towards breastfeeding. The study protocol was approved by the Ethics Committee of the Medical University of Warsaw. To ensure a confidence level of 95%, the minimal sample size was determined as 135 participants, assuming that 4% of pregnant women declare exclusive breastfeeding at 6 months, and predicting a response rate of 80%.

Participants were randomly chosen pregnant Polish women who attended a prenatal appointment at a public outpatient maternity clinic or antenatal course between June and November 2018 in a public university hospital in Warsaw. Women were excluded from the study if they were under 18 years of age, if they experienced pregnancy complications or if they had been educated in the fields of medicine or health sciences (medical doctors, nurses, midwives, dieticians, or laboratory diagnosticians). Data were collected using an original questionnaire designed based on the review of the literature and focus discussion among authors. An anonymous questionnaire was initially piloted on 12 women to evaluate the adequacy of the study tool, the clarity of the questions, time consumption and availability of the needed data. The obtained results were not taken into consideration.

The questionnaire consisted of three parts. Part one assessed socio-demographic and obstetric information which consisted of age, place of living, education, number of children, stage of pregnancy, breastfeeding experience and cord blood storage (Table 1).

In part two, women were asked about their knowledge of stem cells, i.e., their sources, types and their potential use in medicine (Table 2). The maximum knowledge score that could be obtained by the study participant was 18 (one point for the right answer) so that a higher score reflected greater familiarity with stem cells. The women were also asked to point out the main sources of their knowledge about stem cells. Women were given the possibility of indicating more than one option from a list: television/newspapers, internet, medical doctor, nurse/midwife, prenatal courses and family/friends. The instrument also included two items on perceiving and intention to banking and using stem cells in therapy. Finally, women’s positive or negative evaluation of stem cells perception was monitored. All questions in this section were closed-ended, with instructions for participants to choose answers, as appropriate.

Part three of the survey assessed participants’ attitudes toward milk sharing and willingness to overcome problems in breastfeeding practice in the context of information about stem cell content in maternal milk. There were six items (Table 3). Attitudes were measured using a 4-point Likert scale response from 4 = “strongly agree” to 1 = “strongly disagree”. The possible score for the attitudes thereby ranged from 5 to 20, with higher scores reflecting a more positive attitude towards breastfeeding. The last question was about the respondents’ opinions on who should promote information about breast milk stem cells and where it should be promoted. Multiple choice question has allowed respondents to select one or several answers from the list: healthcare practitioners (paediatricians and neonatologist, general practitioner, gynaecologists and obstetricians), antenatal classes, nurses and midwives, the mass media including television, newspapers and the internet.

Data were collected using self-completed questionnaires. The questionnaires were anonymous and coded. Participants consented to participate in the research project. Before providing their answers, the participants were informed about the aim of the study. The questionnaire took approximately 15 min to complete. A total of 165 women were approached to fill in the questionnaire. Out of 165 women, 12 submitted incomplete forms. Three women with education in the fields of medicine or health sciences were excluded from the study. Finally, 150 completed questionnaires were analyzed.

Frequency tables were used to summarize the responses. Univariate analysis was used to describe the characteristics of the pregnant women, as well as knowledge about stem cells and attitudes regarding breastfeeding and preservation of stem cells. Descriptive statistics for the socio-demographic and obstetric characteristics of study participants are presented as frequency and percentage, only the age of respondents is presented as means ±SD. The questions on awareness, attitude, self-rated knowledge and sources of information were analyzed separately, and the responses were reported as counts and percentages. Questionnaire responses were assigned tailed scores for further analysis using non-parametric statistical tests. A Mann–Whitney test was used to chart comparisons of non-normal continuous variables. In the comparative analysis, Pearson’s chi-squared test was applied. The correlation was assessed with the rho-Spearman coefficient. The associations of socio-demographic and obstetric characteristics of women and their knowledge about stem cells, as well as attitudes towards continuing breastfeeding based on awareness of stem cells in breastmilk, were assessed by multiple logistic models. Odds ratios (OR) and their respective 95% confidence intervals (CI) were calculated. The statistical significance level was defined with *p*-values <0.05. The statistical analyses were performed using SPSS Statistics version 25.

## 3. Results

### 3.1. Characteristics of the Study Population

The demographics of the participants are shown in Table 1. The mean age of the pregnant women was 29.84 ± 7.21 years and 80.7% of study participants were city residents. Over two-thirds of respondents graduated from university with at least a Bachelor’s diploma. The women were mostly in their non-first pregnancy (75.3%), with approximately one-half of them having negative breastfeeding experience (52%). Two mothers had a personal experience of cord blood preservation.

### 3.2. Knowledge and Perception of Stem Cells

Among the respondents, almost one-third (61.3%) self-assessed that they had moderate knowledge about stem cells, 2.7% assessed it as high, whereas 30% assessed it as low, only 6% claimed that they did not know anything about stem cells. Responses to the stem cells knowledge items are shown in Table 2. On average, respondents were able to answer correctly about 9 out of 18 knowledge-related questions (mean 8.94 ± 2.5). Less than one-third (28.7%, n = 43) of respondents had a high level of knowledge about stem cells, less than one-quarter realized that breast milk contains stem cells. Umbilical cord blood was the most commonly indicated source of stem cells (78.7%, *p* < 0.001).

The leading source of information regarding stem cells were the mass media including the internet (62.7%), nurses and midwives (29.3%), family and friends (22%), antenatal classes (15.3%) and healthcare practitioners (physicians or gynaecologists and obstetricians—6.7%). Respondents’ positive perception of stem cells as unique and exclusive (96.7%) was not dependant on age, educational level, and stem cell knowledge, even if they had doubts about using stem cells in self/family treatment. In addition, 80% of the study participants would donate their stem cells to be stored in a public or private bank. As far as additional questions, the majority of the studied population showed a good attitude towards stem cell application in a medical setting. 82.7% of women declared they would use the stem cells in therapy for themselves or family members.

Table 3 provides multiple logistic regression analysis used to explore the association of socio-demographic and obstetric characteristics with the knowledge towards stem cells among the study population. The variables were input into the model in order of strength of their association with the knowledge about stem cells as per the simple logistic analysis. The variables included in the final model were based on the degree of importance or relevance of this independent variable with the main outcome. Factors associated with knowledge regarding stem cells were as follows: age, residency status, educational level, number of children, stage of pregnancy. The result of this analysis indicated that residence status, education level, number of children and stage of pregnancy were significantly associated with general knowledge regarding stem cells. University graduates were 5.2 times more likely to have knowledge about stem cells compared to those who finished basic vocational school (OR 5.23, 95% CL 3.27, 14.52). The dominant factor that influenced stem cells knowledge was the stage of pregnancy. The second and third trimester of pregnancy were associated with the highest stem cell knowledge (OR 2.18, 95% CL 1.89, 4.62 and OR 6.45, 95% CL 2.57, 12.85, respectively). Having at least one child increased respondents’ chance for better knowledge compared to childless women (OR 2.65, 95% CL 1.06, 3.56).

### 3.3. Attitudes towards Breastfeeding

Part three of the survey assessed respondents’ attitudes regarding breastfeeding and the existence of stem cells in breast milk. The study participants pointed out that breast milk gives extra benefits apart from nourishing infants (69.6%); it can be used as a unique medicine for premature newborns (68%); breast milk stem cells offer long-term protection against contemporary diseases (69%) (Table 4). We found that 61.3 % of the study participants said that being aware of the presence of stem cells in breast milk could influence the decision to begin or continue breastfeeding. On the other hand, only 36.6% of the study participants declared that they would donate excess milk to a breast milk bank. The study participants indicated who should be the leading source of information regarding breast milk stem cells. Women expected education and support from healthcare practitioners (86.7%), antenatal classes (63.3%), nurses and midwives (60.7%), and in the last place, the mass media including the internet (50%).

The association between socio-demographic and obstetric characteristics of respondents and their attitude towards beginning or continuing breastfeeding based on awareness of stem cells in breastmilk is presented in Table 5. Having at least one child increased participants’ positive attitude towards the continuation of breastfeeding. Similarly, women with a university degree had a significantly more positive attitude regarding continuing breastfeeding (83.7% strongly agree and agree, *p* < 0.001). Among participants who strongly agreed to continue breastfeeding based on the presence of stem cells in breast milk, were women in more advanced stages of pregnancy (38% second and 40% third trimester, *p* = 0.002). The majority of mothers with earlier negative breastfeeding experience claimed they would have a positive attitude towards breastfeeding if they had awareness of stem cells in breastmilk (64%). However, factors that discourage breastfeeding and the duration of breastfeeding were not included in the questionnaire.

Results from the multiple logistic regression analysis examining socio-economic and obstetric characteristics of respondents and their attitudes toward the continuation of breastfeeding based on the presence of breast milk stem cells are shown in Table 6. University education, number of children, stage of pregnancy, breastfeeding experience as well as additional knowledge remained significantly associated with women’s attitudes toward the continuation of breastfeeding based on the awareness of the presence of stem cells in breast milk. Having a child has a statistically significant (*p* = 0.032) correlation with positive attitudes. Respondents with a university degree had a significantly higher probability of considering overcoming difficulties in breastfeeding if they had information about breast milk stem cells compared to those with high education that did not have this knowledge (strongly agree OR 4.56, CI 2.34–6.89; agree OR 21.23, CI 6.45–46.4). Women in more advanced stages of pregnancy were also more likely to continue breastfeeding based on the presence of stem cells in the breast milk (OR 5.46, CL 2.54–11.6 in the second and OR 6.54, CI 1.98–7.58 in the third trimester). Participants with lower stem cells knowledge scores were less likely to consider the continuation of breastfeeding (OR 0.71, CI 0.18–2.83).

## 4. Discussion

Support and consent for stem cell research and therapy have increased in many countries over the past decade. Hence, perinatal stem cell banking is gaining importance especially with the support of government initiatives, which facilitates the growth of knowledge and raises the level of awareness about stem cell storage. Our study shows that the participants had a positive perception of stem cells as unique and exclusive regardless of their age, education level, and detailed knowledge about stem cells. Many of them would have no doubt about using stem cells in self/family treatment. Cord blood is one of the best known safe sources of stem cells. In Poland, that phenomenon can be connected with advertising and a strong private network of cord blood banks, which store cord blood for future use by the donor or his/her relatives. However, many studies have also demonstrated a positive attitude towards cord blood donation, even towards public storage [30,31]. The results obtained show that some pregnant women knew that additional perinatal tissues containing stem cells with potential therapeutic value are routinely discarded as medical waste.

In our study over one-third of the participants reported none or poor prior knowledge about stem cells. However, when we examined knowledge, it was found that 71.3% of respondents had poor knowledge. Our survey may suggest that the views of the study participants on stem cell therapy are framed by beliefs about its benefits and not by well-grounded knowledge. In the literature, there are many papers focusing on cord blood stem cells storage and donation, but there are few on general knowledge of stem cell [32,33,34].

Near two-thirds of the participants (62.7%) reported mass media as the primary source of information about stem cells, whereas in their opinion it should be the healthcare provider (medical doctor, nurse/ midwife). This was not surprising, because other studies show that the main source of health information is the media, including the internet [35,36]. However, we have not found any studies in medical databases dedicated to respondents’ knowledge of stem cells from breast milk. Human breast milk, in addition to nutrients and biologically active substances such as immunoglobulins, growth factors and cytokines, contains a heterogeneous population of cells including stem cells with undefined physiological roles, but strong positive health implications [37]. Hence, there are many medical uses for breast milk, and hospitals use it in treatment plans for many types of patients [38]. In the opinion of researchers and medical doctors, given their ability to reduce inflammation and tissue damage, as well as their multilineage differentiation potential and easy accessibility, breast milk stem cells can prove themselves as an important tool for treating neonatal diseases. In line with our results, women are convinced that breast milk can be used as a unique medicine for premature newborns. It was noted that less than a quarter of the study participants indicated knowledge of the presence of stem cells in breast milk, while the first report on human breast milk stem cells (hBSCs) and new data on the properties and origin of these cells was published in 2012 [39]. The mother’s own milk is the gold standard for the feeding and nutrition of preterm and full-term infants but in situations when premature newborns cannot receive the milk of their biological mothers, donor human milk obtained from well-established human milk banks has become the standard way of feeding [40]. The number of human milk banks has increased in many countries recently [41]. Yet, despite this strong overall understanding, most women would not choose to share milk and donate excess breast milk to milk banks.

Most mothers are knowledgeable when it comes to the benefits of breast milk and breastfeeding. Even though breastfeeding is known to be one of the most effective ways of ensuring child health and survival, the falling rates of breastfeeding and exclusive breastfeeding in many countries is alarming. A study by Krolak-Olejnik from 2017 showed that 97% of Polish mothers started breastfeeding, but unfortunately more than half of newborns are supplemented/bottle-fed even during their stay in the hospital. Only 43.3% continued breastfeeding until the baby was 2 months old [42]. The latest data indicate that breastfeeding rates in Poland have declined dramatically.

At present, midwives provide perinatal education, breastfeeding information and consultation. Besides instrumental support, positive feeding attitudes, the woman’s personal values and beliefs influence her breastfeeding success [43]. Health professionals should empower and broaden knowledge about breastfeeding; however, promoting breastfeeding simply as “normal” makes it likely that mothers will not listen attentively [44]. Therefore, it seems reasonable to look for alternative ways to get the message across by presenting breastfeeding as unique and exceptional. Highlighting the presence of stem cells in breast milk helps to encourage breastfeeding as a non-ordinary activity. Portraying breastfeeding in the media, educational materials, or even breastfeeding support programs as a unique, valuable experience can increase the effectiveness of breastfeeding promotion. Lactation professionals play an important role in convincing mothers to breastfeed. Future interventions to promote breastfeeding should adopt a much broader approach. Health care professionals must know how changing demographics have affected breastfeeding and must adopt a new approach to breastfeeding promotion and support. It is important to keep in mind that women who give birth in the 2020s belong to a new, digital generation [45]. Health promotion, including breastfeeding, should focus on the habits and specific communication preferences of Millennials (born between 1981 and 1996) and Generation Z (people born after 1997), as well as their patterns of gathering and processing information in the digital age. The message should be short, easy to remember and attention-grabbing. Most of all, it should be eye-catching so that the headlines and advertising slogans reach the audience. In our opinion, information about breast stem cells meets these requirements. If we promote breastfeeding using each generation’s communication preferences, we will have a much better chance of increasing breastfeeding rates. It is not enough to implement a breastfeeding education program. It has to be developed using an appropriate strategy, including each generation’s communication preferences and designed in such a way as to help to draw their attention (generational marketing). On the one hand, young generations are more open and seek knowledge, while on the other hand, they have unlimited access to all sorts of information, often unsupported by evidence-based research, but available in an attractive form.

The presence of stem cells in human milk poses numerous questions and implications for breastfeeding, newborns, and maternal health, but also opens a new perspective of future potential applications of these cells in personal and regenerative medicine. Without any doubt, information and knowledge about stem cells in breast milk can make women pay more attention to the benefits of mother’s milk and breastfeeding.

There are a number of limitations to this study. First, study participants were recruited in only one city and one hospital in Poland and hence the findings may not be generalizable to other populations. However, there is no reason to believe that these participants differ significantly from other women. Factors that discourage breastfeeding and the stages of stopping lactation in previous experience were not included in the questionnaire. We did not ask about religious affiliation, which can be an important basis for disapproval of stem cells. In the questionnaire, the question about the cord blood bank did not distinguish between public and private stem cell bank and the participant could not make a choice between public or personal storage. In Poland, a public-private model of stem cell banking exists, however, the donation of cord blood for public banking is marginal. Private banking is preferentially promoted, aggressively advertised, and used almost exclusively by the wealthier population. Hence the positive perception of stem cells may arise in the population. On the other hand, the donation of tissue for public banking is considered a gift of moral and social value, which also can encourage the donation of stem cells. Additional studies with a larger sample size of women and women from other countries are needed to further characterize attitudes towards breastfeeding associated with breast milk stem cell knowledge.

## 5. Conclusions

The survey results indicated that city residence, university degree, maternal experience and advanced pregnancy correlated with higher general knowledge regarding stem cells. In this original research study, pregnant women had a positive perception of stem cells as unique and exclusive and declared that the information about stem cells in breast milk could influence the decision to continue and extend the time of breastfeeding. Hence, highlighting the presence of stem cells in breast milk by healthcare practitioners and lactation professionals can help to encourage breastfeeding. Individual, personal factors but also knowledge about breast milk stem cells may be of particular importance in overcoming difficulties in feeding a child in a situation of limited access to professional lactation support.

## Figures and Tables

**Table 1 ijerph-18-02382-t001:** Socio-demographic and obstetric characteristics of pregnant women responding to the survey about stem cells (n = 150).

Variable	Frequency/(Percent)
Age, Mean ± SD	29.84 ± 7.21
City resident	
Yes	121 (80.7)
No	29 (19.3)
Education level	
Bachelor’s degree or higher	106 (70.7)
High school	27 (18)
Basic vocational school	17 (11.3)
Number of children	
0	37 (24.7)
1	53 (35.3)
2	55 (36.7)
≥3	5 (3.3)
Stage of pregnancy	
First trimester	56 (37.3)
Second trimester	49 (32.7)
Third trimester	45 (30)
Breastfeeding experience	
No	42 (28)
Yes—positive	30 (20)
Yes—negative	78 (52)
Cord blood storage	
No	148 (98.7)
Yes	2 (1.3)

**Table 2 ijerph-18-02382-t002:** Knowledge and perception of stem cells by pregnant women (n = 150).

Statement	Possible Answers	Frequency/Percent
Do you have knowledge of stem cells?	No	9 (6)
	Yes	141 (94)
	Low	45 (30)
	Moderate	92 (61.3)
	High	4 (2.7)
General stem cell knowledge (correct answer in parentheses) ^a^	Stem cells are unspecialized (T)	59 (39.3)
	Stem cells are capable of dividing and self-renewal (T)	71 (47.3)
	Stem cells can generate one or more specialized cell types (T)	84 (56)
	Stem cells are involved in tissue regeneration (T)	48 (32)
	Stem cells are abundant in the foetus (T)	118 (78.7)
	There are no stem cells in adult humans (F)	55 (36.7)
	Stem cells are present in cord blood (T)	120 (80)
	There are no differences between stem cells and body cells (F)	5 (3.3)
	Stem cells are harvestable during a safe and minimally invasive procedure (T)	90 (60)
	All stem cell research and therapy are extremely controversial (F)	30 (20)
	Breast milk contains unique stem cells (T)	34 (22.7)
Your sources of general information about stem cells.	Television, newspapers (including magazines for pregnant women and mothers)	6 (4)
	Internet	88 (58.7)
	Medical doctor: physician or gynaecologist and obstetrician	10 (6.7)
	Nurse/ midwife	44 (29.3)
	Prenatal courses	23 (15.3)
	Family and friends	33 (22)
Indicate safe sources of stem cells for medical treatments (correct answer in parentheses) ^a^	Perinatal tissue like placental, amniotic fluid, excluding cord blood (T)	36 (24)
	Cord blood (T)	118 (78.7)
	Foetus (F)	0
	Blood marrow (T)	90 (60)
	Skin (T)	9 (6)
	Adipose tissue (T)	25 (16.7)
	Breast milk (F)	34 (22.7)
How do you perceive stem cells?	Negative and controversial	5 (3.3)
	Positive and exclusive	145 (96.7)
Would you like to preserve perinatal stem cells in a public/private bank?	Yes	120 (80)
	No	14 (9.3)
	I don’t know about this possibility	16 (10.7)
Are you agree with using stem cells in self/family treatment.	Yes	124 (82.7)
	No	4 (2.7)
	I don’t know about this possibility	22 (14.6)

^a^ T = True, F = False.

**Table 3 ijerph-18-02382-t003:** Multiple logistic regression analysis of socio-demographic and obstetric characteristics of respondents and their general knowledge regarding stem cells (n = 150).

	Knowledge about Stem Cells			
Variable	High n = 43	Low n = 107	*p*-Value	OR (95% CI)
Age, Mean ±SD	34.2 ± 5.8	26.4 ± 7.2	<0.001	1.02 (099, 1.09)
**City resident,**	N (%)	N (%)	0.024	
Yes	22 (51)	99 (93)		1.00
No	21 (49)	8 (7)		1.02 (0.96, 1.5)
**Education level,**	n(%)	n(%)	<0.001	
Bachelor’s degree or higher	41(95)	66 (61)		5.23 (3.27, 14.52)
High school	1 (2.5)	26 (24)		2.73 (1.98, 4.78)
Basic vocational school	1 (2.5)	17 (16)		1.00
**Number of children,**	n(%)	n(%)	0.01	
0	5 (11.6)	32 (30)		1.00
1	14 (32.6)	39 (36.5)		2.65 (1.06, 3.56)
2	20 (46.5)	35 (32.6)		3.56 (2.56, 4.49)
≥3	4 (9.3)	1 (0.9)		1.63 (0.85, 3.23)
**Stage of pregnancy**			<0.001	
First trimester	2 (4.6)	54 (50.5)		1.00
Second trimester	5 (11.6)	44 (41)		2.18 (1.89, 4.62)
Third trimester	36(84.7)	9 (8.4)		6.45 (2.57, 12.85)
**Breastfeeding experience**			NS	
No	20 (46.5)	22 (20.6)		1.00
Yes (total)	23 (52.5)	85 (79.4)		−0.38 (0.17, 0.94)
positive	16 (37.2)	14 (13)		-
negative	7 (16.3)	71 (66.4)		-

**Table 4 ijerph-18-02382-t004:** Assessment of attitudes towards stem cells including breast milk stem cells among Polish pregnant women.

Statement	Answer			
	Strongly Agree (%)	Agree (%)	Disagree (%)	Strongly Disagree (%)
Breast milk has extra benefits beyond nourishing infants and this is connected with the presence of stem cells.	70 (46.6)	33 (22)	43 (28.7)	4 (2.7)
Breast milk stem cells may be related to the positive long-term effect of breastfeeding and the prevention of diseases of modern life.	17 (11.3)	87 (58)	41 (27.3)	5 (3.3)
Awareness of the presence of stem cells in breast milk could influence the decision to begin/continue/extend the time of breastfeeding.	50 (33.3)	42 (28)	38 (25.3)	20 (13.3)
Information about stem cells in breast milk can influence your decision to donate excess milk to a breast milk bank.	5 (3.3)	50 (33.3)	30 (20)	65 (43.3)
Breast milk can be used as a medicine that no one else can make for premature newborns.	26 (17.3)	76 (50.7)	26 (17.3)	22 (14.6)
Who should promote information about the presence of stem cells in human milk?		Frequency/percent		
Medical doctors paediatricians and neonatologists general practitioner gynaecologists and obstetricians		130 (86.7) 47 (31.3) 41 (27.3) 42 (28)		
Nurses/midwives		91 (60.7)		
Antenatal classes		95 (63.3)		
Television, newspapers		15 (10)		
Internet and social media		60 (40)		

**Table 5 ijerph-18-02382-t005:** Association between socio-demographic and obstetric characteristics of respondents and their attitude to begin/continue/extend breastfeeding based on awareness of stem cells in breast milk.

	Attitude to the Continuation of Breastfeeding Based on The Presence of Breast Milk Stem Cells				
Variable	Strongly Agree n = 50	Agree n = 42	Disagree n = 38	Strongly Disagree n = 20	*p*-Value
**Age**, Mean ±SD	30.2 ± 6.8	29.7 ± 5.4	28 ± 7.1	32 ± 4.2	NS
**City resident**, n(%)					NS
Yes	41 (82)	34 (81)	31 (81.6)	15 (75)	
No	9 (18)	8 (19)	7 (18.4)	5 (25)	
**Education level,** n(%)					<0.001
Bachelor’s degree or higher	43 (86)	34 (81)	22 (58)	7 (35)	
High school	6 (12)	6 (14)	8 (21)	7 (35)	
Basic vocational school	1 (2)	2 (5)	8 (21)	6 (30)	
**Number of children,**					0.032
0	11 (22)	17 (40)	5 (13)	4 (20)	
1	22 (44)	10 (24)	15 (39)	6 (30)	
2	17 (34)	13 (31)	16 (42)	9 (45)	
≥3	0 (0)	2 (5)	2 (5)	1 (5)	
**Stage of pregnancy**					0.002
First trimester	11 (22)	23 (55)	14 (36)	8 (40)	
Second trimester	19 (38)	11 (26)	12 (32)	7 (35)	
Third trimester	20 (40)	8 (19)	12 (32)	5 (10)	
**Breastfeeding experience**					0.004
No	21 (42)	14 (33.3)	5 (13)	2 (10)	
Yes (total)	29 (58)	28 (66.7)	33 (87)	18 (90)	
Positive	2 (4)	5 (12)	13 (34)	10 (50)	
Negative	27 (54)	23 (54.7)	20 (53)	8 (40)	

**Table 6 ijerph-18-02382-t006:** Multiple logistic regression analysis between socio-economic and obstetric characteristics of respondents and their attitudes toward the continuation of breastfeeding based on the presence of breast milk stem cells.

	OR (95% CI)			
Variable	Strongly Agree	Agree	Disagree	Strongly Disagree
**Age**, Mean ±SD	0.91 (0.38, 2.14)	2.54 (0.93, 6.94)	1.63 (0.64, 4.11)	1.05 (0.92, 1.09)
**City resident**				
Yes	1.00	1.00	1.00	1.00
No	0.93 (0.27, 2.12)	1.11 (0.35, 3.5)	1.32 (0.32, 4.76)	0.47 (0.21, 1.04)
**Education level**				
Bachelor’s degree or higher	4.56 (2.34, 6.89)	21.23 (6.45, 46.4)	3.19 (0.73, 12.38)	2.98 (1.01, 8.70)
High school	1.00	1.00	1.00	1.00
Basic vocational school	-	-	-	-
**Number of children**				
0	1.00	1.00	1.00	1.00
1	1.98 (0.89, 2.4)	1.78 (0.67, 7.21)	0.65 (0.26, 1.59)	1.60 (0.71, 4.98)
2	2.98 (1.87, 3.78)	1.96 (0.71, 5.15)	0.65 (0.26, 1.59)	1.95 (0.54, 6.32)
≥3	-	-	-	-
**Stage of pregnancy**				
First trimester	1.00	1.00	1.00	1.00
Second trimester	5.46 (2.54, 11.6)	4.89 (2.35, 7.25)	1.57 (0.98, 1.98)	1.18 (0.79, 3.12)
Third trimester	6.54 (1.98, 7.58)	9.54 (3.56, 12.78)	0.97 (0.38, 2.34)	1.12 (0.79, 3.62)
**Breastfeeding experience**				
No	1.00	1.00	1.00	1.00
Yes	3.56 (2.56, 7.34)	2.78 (1.09, 7.68)	2.04 (0.78, 5.47)	1.24 (0.63, 1.98)
**Knowledge score**				
Score up to 10 points	1.00	1.00	1.00	1.00
Score ≤ 9 points	0.71 (0.18, 2.83)	1.11 (0.58, 5.83)	3.25 (1.07, 8.05)	2.98 (1.89, 6.88)

## Data Availability

Derived data supporting the findings of this study are available from the corresponding author (M.W.Z.) on request.

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
