# Peer review of "How Knowledge about Stem Cells Influences Attitudes towards Breastfeeding: Case Study of Polish Women"

_ijerph, 2021, doi:10.3390/ijerph18052382_

Round 1
Reviewer 1 Report
First of all, I would like to thank the authors for choosing a topic as new as stem cells and as important as breastfeeding.
The introduction explains the main advantages of breastfeeding and how important the support of health professionals is to ensure the success of breastfeeding. The benefits of stem cells and their presence in breast milk are then explained.
The aim of this research is to show that if women receive adequate information about the stem cells that are abundant in milk, they will be more likely to breastfeed.
The methodology used for the development of the research seems to me to be correct. There is a clear description of how the sample was selected, the inclusion and exclusion criteria and the instruments used.
The strong point of this research is that multiple logistic regression models were used to associate the different variables, thus providing scientific rigour to the results.
The results measure women's knowledge and perception of stem cells and women's attitudes towards breastfeeding.
The discussion is also well developed, comparing the results of the research with the existing scientific literature and acknowledging the limitations of the study.
The conclusions are supported by the results and highlight the role of health professionals in promoting breastfeeding.
To conclude my assessment, I would like to congratulate the authors for this much improved version of the manuscript compared to the first version.
This time I do not have any negative aspects to comment on as all the recommendations given in the review of the first version of the manuscript have been taken into account.
I can only say that a good job has been done.
Congratulations!
Thank you very much.
Best regards
Author Response
Point 1: This time I do not have any negative aspects to comment on as all the recommendations given in the review of the first version of the manuscript have been taken into account.
Response 1:
Dear Reviewer,
thank you for your nice opinion. We really appreciate your comments and recommendations which helped us improve the quality of our manuscript.
Thank you very much.
Sincerely
Reviewer 2 Report
The work is carried out exhaustively and scrupulously analyzed
Two remarks:
1) The text reports that over 80% of the participants report declare that they live in a residential area. I suggest that this parameter is not considered in the statistical analysis
2) In the questionnaire, the question about the cord blood bank does not distinguish between public and private banks and the participant cannot make a choice between public and private banking cord blood.
Also, the next question explicitly asks if the participant agrees to the use of stem cells in self / family treatment and this can generate a message in favor of private banking, which should not be promoted.
This element should emerge in the discussion
Author Response
Point 1: The text reports that over 80% of the participants report declare that they live in a residential area. I suggest that this parameter is not considered in the statistical analysis.
Response 1: We note that is a disagreement between the reviewers about the usefulness of the information about residence status, and only Reviewer#2 recommended that information be dropped. Except that, the result of analysis indicated that residence status, next to education level, number of children, and stage of pregnancy were significantly associated with general knowledge regarding stem cells.
We are concerned that omitting this information might contribute to a lack of transparency. Also, in discussion with colleagues on this topic, we believe readers should be informed about it.
Point2: In the questionnaire, the question about the cord blood bank does not distinguish between public and private banks and the participant cannot make a choice between public and private banking cord blood.
Also, the next question explicitly asks if the participant agrees to the use of stem cells in self / family treatment and this can generate a message in favor of private banking, which should not be promoted.
This element should emerge in the discussion.
Response 2: The remark of the Reviewer#2 was included and mark in the text using the "Track Changes" function (page 15, lines: 2-9).
This manuscript is a resubmission of an earlier submission. The following is a list of the peer review reports and author responses from that submission.
Round 1
Reviewer 1 Report
First of all to say that the subject of this manuscript is very interesting. Health professionals should investigate the benefits that breastfeeding has for the mother and her child and try to promote it in our society.
However, I think this manuscript has many things that need to be improved:
1. Introduction: In content it seems complete, but there are many statements that have no reference. For example:
- line 32 "Despite.....century "
- line 39 "Some European... since then"
- line 42 "According....weeks.
- line 52 "There....rates"
- line 54 "However...decisions"
- line 61 "due to...discussions"
- line 67 " Hence, ....therapies"
- line 74 "From....cord"
- line 78 "Confirmation....breastfeeding
2. Matherial and Methods
It would be necessary to know the reliability of the questionnaire used. Apparently, this is a self-made quiz.
on line 117 it says that the reliability of the questionnaire has been ensured through a pilot test, but the actual reliability of the test is not specified by calculating Cronbach's alpha for each of the domains and for the entire questionnaire.
3. Results
Results are not clearly displayed.
I would have liked to see tables where you can see the comparisons of the different variables, applying Chi square or Mann-Witney test as appropriate in each case.
4. Discussion
As in the introductory section, there are many statements that are not bibliographically referenced.
5. Conclusions
The conclusions must be supported by the results.
We already knew that breastfeeding supports and protects maternal and infant health before writing this article, therefore this would not be a conclusion.
Professionals should promote breastfeeding, but I don't understand why they talk about telephone counseling in times of a Covid pandemic. This has nothing to do with the results.
I regret not being able to give a more favorable assessment of this manuscript. As I said at the beginning, I find the topic very interesting and I am sure that they will be able to make an improved version so that it can be published.
Kind Regards
Reviewer 2 Report
Thank you for giving me an opportunity in reviewing the article. The topic related to stem cell in human breast milk is new. However, my major concern is about the design and hypothesis of this study:
attitude can influent behaviour but not exactly equal to behaviour. I think the finding would be more interesting if the study can look at the correlation between stem cell knowledge and actual breastfeeding behaviour, and find out whether attitude is a mediator. A cross-sectional retrospective study among new mothers might be able to answer the above research question.